# Capsule Network Projectors are Equivariant and Invariant Learners

**Miles Everett**                                                          *miles.everett@abdn.ac.uk*
*Department of Computing Science*
*University of Aberdeen, UK*

**Aiden Durrant**                                                          *aiden.durrant@abdn.ac.uk*
*Department of Computing Science*
*University of Aberdeen, UK*

**Mingjun Zhong**                                                          *mingjun.zhong@abdn.ac.uk*
*Department of Computing Science*
*University of Aberdeen, UK*

**Georgios Leontidis**                                                     *georgios.leontidis@abdn.ac.uk*
*Interdisciplinary Institute*
*Department of Computing Science*
*University of Aberdeen, UK*

**Reviewed on OpenReview:** *https://openreview.net/forum?id=7owCO3qskH*

## Abstract

Learning invariant representations has been the long-standing approach to self-supervised learning. However, recently progress has been made in preserving equivariant properties in representations, yet do so with highly prescribed architectures. In this work, we propose an invariant-equivariant self-supervised architecture that employs Capsule Networks (CapsNets), which have been shown to capture equivariance with respect to novel viewpoints. We demonstrate that the use of CapsNets in equivariant self-supervised architectures achieves improved downstream performance on equivariant tasks with higher efficiency and fewer network parameters. To accommodate the architectural changes of CapsNets, we introduce a new objective function based on entropy minimisation. This approach, which we name CapsIE (**Caps**ule **I**nvariant **E**quivariant Network), achieves state-of-the-art performance on the equivariant rotation tasks on the 3DIEBench dataset compared to prior equivariant SSL methods, while performing competitively against supervised counterparts. Our results demonstrate the ability of CapsNets to learn complex and generalised representations for large-scale, multi-task datasets compared to previous CapsNet benchmarks. *Code is available at https://github.com/AberdeenML/CapsIE.*

## 1 Introduction

Equivariance and invariance have become increasingly important properties and objectives of deep learning in recent times, with precedence being largely placed on the latter. The task of invariance, i.e., being able to classify a specific object regardless of the camera perspective or augmentation applied, has driven progress in modern self-supervised learning approaches, specifically those that follow a joint embedding architecture (Assran et al., 2022; Bardes et al., 2022; Chen et al., 2020). Equivariance, on the other hand, is the task of capturing embeddings which equally reflect the translations applied to the input space in the latent space. Equivariance thus has become an important property to capture to enable the learning of high-quality representations in the real world, where transformations such as viewpoint are essential.

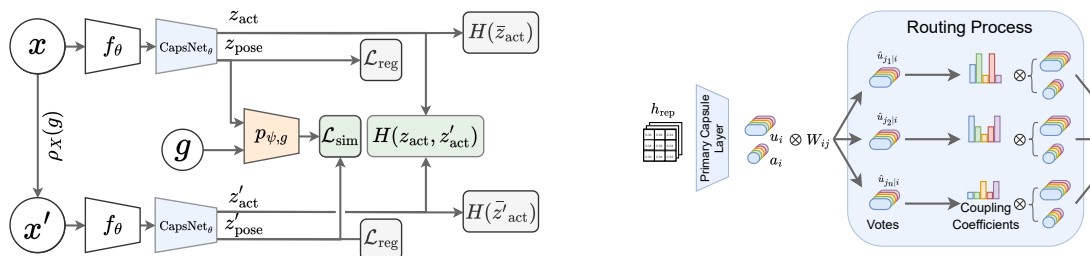

(a) Schematic overview of the CapsIE architecture.    (b) Generalised visualisation of the CapsNet projector.

Figure 1: *Left*: **Schematic overview of the proposed CapsIE architecture.** Representations are fed into a CapsNet projector, and the output embeddings $Z_{\text{act}}$ and $Z_{\text{pose}}$ correspond to invariant and equivariant embeddings, respectively. *Right*: **Generalised view of a Capsule projection head.** CNN feature maps are transformed via the primary capsules into poses $u_i$, represented by cylinders, and activations $a_i$, represented by circles. Poses are transformed to votes, which represent a lower-level capsule's prediction for each of the higher-level capsules. The routing process then determines how well these votes match the concept represented by the upper-level capsule, thereby creating the coupling coefficients. Coupling coefficients inform $u_j$ and $a_j$, the output of the capsule projector head.

Self-supervised learning owes its success to invariant objectives, where all recent progress, whether that is by contrastive (Chen et al., 2020), information-maximisation (Bardes et al., 2022; Zbontar et al., 2021), or clustering-based methods (Caron et al., 2021; Assran et al., 2022) rely on ensuring invariance in their representations under augmentation. This setting ensures performance in classification-based tasks; however, when employing the representations in alternative tasks, preserving information is essential to improve generalisation. To maintain properties of the transformation, one can predict the augmentations applied (Dangovski et al., 2022; Lee et al., 2021), yet this is typically not considered truly equivariant, given that the mapping of transformations is not represented in the latent space. Methods that employ such a prediction methodology are typically considered equivariant, as the transformation in the input space is preserved in the latent space. Here, prediction networks are employed to reconstruct the view prior to transformation (Winter et al., 2022), learn symmetric representations (Park et al., 2022), or predict the latent representation of the transformed view from the representation of the original view given the transformation parameters (Garrido et al., 2023).

The above methods, although promising, enforce equivariance via objective functions on vector representations, yet these methods fail to employ architectural approaches that have been shown to be capable of better capturing these properties. Capsule Networks (CapsNets), which utilise a process called routing (Sabour et al., 2017; Hinton et al., 2018; Everett et al., 2023; Hahn et al., 2019; De Sousa Ribeiro et al., 2020; Liu et al., 2024), are one such architecture, exhibiting desirable properties that other state-of-the-art (SOTA) architectures, such as Vision Transformers (ViTs) and CNNs, lack. Specifically, CapsNets have demonstrated a natural ability to possess strong viewpoint equivariance and viewpoint invariance properties. They achieve this through their ability to capture equivariance with respect to viewpoints in neural activities, and invariance in the weights. In addition, viewpoint changes have nonlinear effects on pixels but linear effects on object relationships (De Sousa Ribeiro et al., 2020; Hinton et al., 2018). Ideally, these properties could lead to the development of more sample-efficient models that can exploit robust representations to better generalise to unseen cases and new samples.

However, a common argument is that CapsNets have only shown these properties on toy examples such as the smallNORB dataset (LeCun et al., 2004), which many would consider irrelevant for modern architectures. Despite this, small CapsNets outperform much larger CNN and ViT counterparts (Everett et al., 2023). In this work, we propose a novel CapsNet formulation and corresponding objective function, achieving SOTA on equivariant viewpoint rotation tasks on the 3DIEBench dataset (Garrido et al., 2023), which has been created to specifically benchmark equivariant and invariant properties of deep learning models. We prove that CapsNets retain their desirable properties on this dataset, which is considerably more difficult than what has been previously achieved with CapsNets, while also establishing new SOTA for this dataset.

To summarise, our contributions are:

- We propose a novel architecture (Figure 1) based on a Capsule Network projection head that utilises the key assumptions of capsule architectures to learn equivariant and invariant representations, which does not require the explicit split of representations.

- We design a new objective function to accommodate the employment of a CapsNet projector, enforcing invariance through entropy minimisation.

- We show state-of-the-art performance on 3DIEBench classification for equivariance benchmark tasks from our CapsNet-based architecture.

## 2 Problem Statement

Typically, self-supervised learning maximises the similarity between embeddings of two augmented views of an image such that they are invariant to augmentations, and instead capture semantically meaningful information of the original image. Views $x$ and $x'$ are each transformed from an image $d \in \mathbb{R}^{c \times h \times w}$ sampled from dataset $\mathcal{D}$ by image augmentations $\tau, \tau' \sim T$ sampled from a set of augmentations $T$. Embeddings are obtained by feeding each view through an encoder $f_\theta$, where the output representations $y, y'$ are fed through a projection head $h_\phi$ to produce embeddings $z, z'$ whose similarity is maximised. However, it is detrimental in many settings that $f$ is invariant to all transformations; instead, in this work, we are focused on ensuring that $f$ is equivariant to viewpoint transformations. To train for and evaluate such properties, we base our study on the challenging 3DIEBench dataset and the corresponding problem definition presented in Garrido et al. (2023).

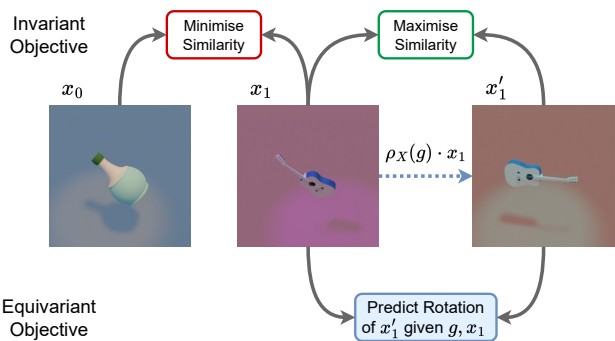

Figure 2: **Visual depiction of the problem statement.** Two images are represented by subscripts $0, 1$ while view under transformation $g$ is given by $'$. Arrows represent the construction of embeddings from an encoder network. ***Top*** the *invariance* objective is to maximise the similarity between embeddings of views originating from the same image. ***Bottom*** the *equivariance* objective aim to learn the transformations $\rho_X(g) \cdot x$ applied to $x$.

As defined in Garrido et al. (2023) in which the dataset was proposed the goal (visually depicted in Figure 2) is to learn an encoder network $f$ and predictor network $P$ to construct representations that are equivariant to viewpoint transformations when the transformation group action $\rho$ on the input is not known, but the group elements $g$ that parameterise the transformations are known. Further details of these transformations and the benchmark 3DIEBench dataset and the group theory defining the problem statement are given in Sections A.1 and A.2 respectively.

## 3 Method

### 3.1 Architecture

Our method, which we name CapsIE (**Caps**ule Network **I**nvariant **E**quivariant), follows the general joint embedding architecture previously described in Section 2 and extends those proposed by VICReg Bardes et al. (2022) and Split Invariant Equivariant (SIE) Garrido et al. (2023) methods. Like previous methods, we employ a ResNet-18 He et al. (2016) encoder as the core feature extractor $f_\theta$ of our network. Yet, unlike SIE Garrido et al. (2023), we do not split the representations, and therefore do not require the use of separate invariant and equivariant projection heads. Instead, we employ a single CapsNet (described in Section 3.2) which takes as input the full representation of the encoder in place of the multi-layer perceptron (MLP) in the projection head $h_\phi$. Given the architectural design of CapsNets, our projection head outputs both an

activation scalar, representing how active the capsule is, and a $4 \times 4$ pose for each capsule. A simplified visual representation can be seen in Figure 3.

To align with the above problem statement and Equation 8, we aim to simultaneously learn invariant and equivariant representations by optimising our network $f$ with respect to the output activations and poses. In this case, we consider the vector of activations to capture the existence of semantic concepts/objects of the input; thus, the invariant information is preserved by the transformation. The pose, on the other hand, is designed to encode positional information related to each corresponding capsule (De Sousa Ribeiro et al., 2024) (i.e. semantic concept), therefore, it contains equivariant information that was changed by the transformation. Akin to the SIE method, we therefore consider two embedding vectors for each image view, $z_{\mathrm{act}}$ and $z_{\mathrm{pose}}$, which correspond to the capsule activation and pose, and invariant and equivariant components, respectively.

To enforce our network to learn equivariant properties we utilise a prediction network $p_{\psi,g}$ which takes as input the transformation $g$ and $z_{\mathrm{pose}}$ to predict $z'_{\mathrm{pose}}$, and hence learn $\rho_Y(g)$. In our setting, $g \in \mathbb{R}^3$ corresponds to Tait-Bryan angles of the rotation applied 6. For prediction and the subsequent evaluation, quarterions are employed in place of the Tait-Bryan angles. In this work, we employ the hypernetwork approach proposed by (Garrido et al., 2023), which uses a linear projector that takes as input the transformation parameters $g$ to parameterise an MLP predictor. Such a network avoids the case where the transformation parameters $g$ are ignored and the predictor provides invariant solutions. We present more details of the predictor network in the appendix, and visually depict the full CapsIE architecture in Figure 1.

## 3.2   Capsule Network Projector

**Capsule Viewpoint Equivariance**   CapsNets are designed to handle spatial hierarchies and recognise objects regardless of their orientation or location, achieving equivariance through their structure (Ribeiro et al., 2020). A capsule is a group of neurons – vector-based representations – representing instantiation parameters such as position, orientation, and size. Before any routing process begins, lower-level capsule poses $u_i$ are transformed to $n$ upper-level capsule poses $u_{j|i}$ which align with concepts represented by higher-level capsules, preserving spatial relationships and hierarchical information. It is then determined through the routing process how well these transformed poses correspond with the concept represented by the upper-level capsule. CapsNets, unlike convolution, excel in achieving viewpoint invariance and viewpoint equivariance as they can capture equivariance with respect to viewpoints in neural activities, and invariance in the network's weights (De Sousa Ribeiro et al., 2024). Consequently, capsule routing aims to detect objects by looking for agreement between their parts, thereby performing equivariant inference.

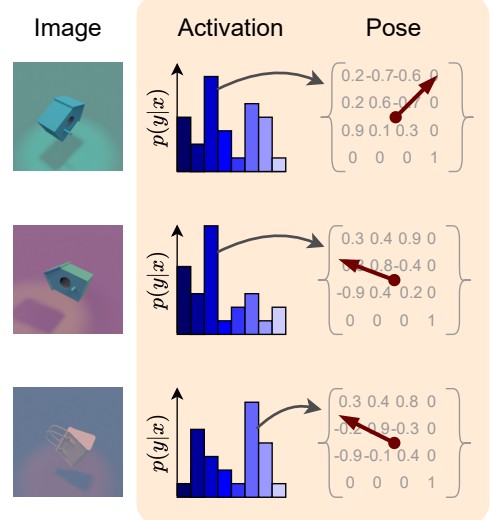

Figure 3: **Simplified visual representation of CapsNet outputs.** Vector of activations outputs the probability of each capsule being activated, whereas the pose matrix corresponds to the object pose in relation to the frame.

**Self Routing Capsule**   We use the Self Routing CapsNet (SRCaps) (Hahn et al., 2019) based on the efficiency of its non-iterative routing algorithm. We consider the trade-off of a small amount of classification accuracy to be acceptable when comparing the performance of SRCaps to other capsule architectures, which require significantly more resources to train. Based on the size of the 3DIEBench dataset, these other routing algorithms would be unsuitable.

SRCaps calculates the coupling coefficients between each capsule in lower layer $i$ with each capsule in upper layer $j$ to produce the coupling coefficients $c_{ij}$. It does so by using a learnable routing matrix $W^{route}$ multiplied with the lower capsule pose vector $u_i$, mimicking a single-layer perceptron to produce routing coefficients $b_{ij}$ which when passed through a softmax function produce coupling coefficients $c_{ij}$. Additionally,

we determine the activation of upper-level capsules $a_j$ by first multiplying $a_i$ by $c_{ij}$ to create votes and then dividing this by $a_i$ to create weighted votes.

$$c_{ij} = \text{softmax}(W_i^{route} u_i)_j, \quad a_j = \frac{\sum_{i \in \Omega_l} c_{ij} a_i}{\sum_{i \in \Omega_l} a_i} \tag{1}$$

The output pose of a capsule layer is calculated using a learnable weight matrix $W^{pose}$ which when multiplied with $u_i$ provides a capsule pose of each lower-level capsule for each upper-layer capsule i.e $u_{j|i}$. Following the same procedure as the activations, $u_j$ is the weighted sum of these poses by $a_j$.

$$\hat{u}_{j|i} = W_{ij}^{\text{pose}} u_i, \quad u_j = \frac{\sum_{i \in \Omega_l} c_{ij} a_i \hat{u}_{j|i}}{\sum_{i \in \Omega_l} c_{ij} a_i}. \tag{2}$$

### 3.3 Objective Functions

**Invariant Criterion.** To train our aforementioned architecture, we first introduce an invariant objective as the cross entropy between activation probability vectors, $H(Z_{\text{act}}, Z'_{\text{act}})$ where $Z$ refers to the matrix embeddings over a batch. The aim is to enforce embedding probability pairs originating from the same image to be matched. To avoid trivial solutions and collapse to a single capsule, we employ the mean entropy maximisation regularisation (Assran et al., 2021; 2022) on the same activation probability vectors to encourage the model to utilise the full set of capsules over a batch. This regularisation maximises the entropy of the mean probabilities $H(\bar{Z}_{\text{act}})$ and $H(\bar{Z}'_{\text{act}})$, where $\bar{Z}_{\text{act}} = \frac{1}{B} \sum_{i=1}^{B} Z_{\text{act}}$ and $B$ is the batch size.

**Equivariant Criterion.** As previously stated in Section 2, our goal is to learn the predictor $p_{\psi,g}$ to model $\rho_Y(g)$ as to enforce equivariant representations. This is achieved by minimising a $L2$ (Euclidean Distance) objective between the output vector of the predictor $p_{\psi,g}(Z_{\text{pose}})$ given translation parameters $g$ and equivariant representation $Z_{\text{pose}}$, and the augmented view's equivariant representation vector $Z'_{\text{pose}}$. To avoid collapse and improve training stability we also regularise the output of $p_{\psi,g}(Z_{\text{pose}})$ by ensuring the variance of the predicted equivariance representation is 1 to avoid collapse. Whereas, SIE (Garrido et al., 2023) finds this to be an optional but recommended component, we found in practice, without such regularisation the predictor would consistently collapse to trivial solutions.

As with the activation vector, we employ variance–covariance regularisation on the pose to prevent the representations from collapsing into trivial solutions. The variance objective $V$ ensures that all dimensions $d$ in the embedding vector are equally utilised while the covariance objective $C$ decorrelates the dimensions to reduce redundancy across dimensions. The regularisation for equivariant vectors $\mathcal{L}_{\text{reg}}$ is given by

$$\mathcal{L}_{\text{reg}}(Z) = \lambda_C \, C(Z) + \lambda_V \, V(Z), \quad \text{where} \tag{3}$$

$$C(Z) = \frac{1}{d} \sum_{i \neq j} Cov(Z)_{i,j}^2 \quad \text{and} \quad V(Z) = \frac{1}{d} \sum_{j=1}^{d} \max \left( 0, 1 - \sqrt{Var(Z_{\cdot,j})} \right). \tag{4}$$

The final objective function is given by the weighted sum of the individual objectives:

$$\mathcal{L}(Z_{\text{act}}, Z'_{\text{act}}, Z_{\text{pose}}, Z'_{\text{pose}}) = \lambda_{\text{inv}} H(Z_{\text{act}}, Z'_{\text{act}}) - (H(\bar{Z}_{\text{act}}) + H(\bar{Z}'_{\text{act}})) + \tag{5}$$

$$\lambda_{\text{equi}} \frac{1}{N} \sum_{i=1}^{N} \| p_{\psi,g_i}(Z_{i,\text{pose}}) - Z'_{i,\text{pose}} \|_2^2 + \tag{6}$$

$$\mathcal{L}_{\text{reg}}(Z_{\text{pose}}) + \mathcal{L}_{\text{reg}}(Z'_{\text{pose}}) + \lambda_V V(p_{\psi,g_i}(Z_{i,\text{pose}})). \tag{7}$$

## 4 Experimentation

### 4.1 Training Protocol

To directly compare with prior works employing the 3DIEBench dataset, we follow an identical training protocol, as defined in Garrido et al. (2023). All methods employ a ResNet-18 encoder network ($f_\theta$). For the

Table 1: **Evaluation of invariant properties via downstream classification task.** Representations are learnt under the invariance and rotation equivariant objective. We evaluate both the representations and the intermediate embeddings of the projection head under varying numbers of capsules. FLOPs and Parameters correspond to computation during training, *'-' refers to non-compatible experiments*. For non-capsule models, 'All', 'Inv.', and 'Equi.' refer to the full vector representation, the "left" half representation optimised under the invariant objective, and the "right" half representation optimised under the equivariant objective, respectively. For capsule networks, 'Inv.', and 'Equi.' refer to the activation vector and pose matrices, respectively.

| Method | Computational Load | | Embedding Dims | | Classification (Top-1%) | | |
| | Parameters | # FLOPs | Inv. | Equi. | All | Inv. | Equi. |
| --- | --- | --- | --- | --- | --- | --- | --- |
| Supervised | | | | | | | |
| ResNet-18 | 11.2M | 3.09G | - | - | 87.47 | - | - |
| SR-Caps - 16 | 11.0M | 3.16G | - | - | - | 73.85 | - |
| SR-Caps - 32 | 13.0M | 4.27G | - | - | - | 59.70 | - |
| SR-Caps - 64 | 18.7M | 8.22G | - | - | - | 69.45 | - |
| Encoder Representation | | | | | | | |
| SIE | 20.1M | 13.07G | 512 | 512 | 82.94 | 82.08 | 80.32 |
| CapsIE - 16 | 12.7M | 3.49G | 16 | 256 | 76.51 | - | - |
| CapsIE - 32 | 14.7M | 4.57G | 32 | 512 | 79.14 | - | - |
| CapsIE - 64 | 20.4M | 8.69G | 64 | 1024 | 79.60 | - | - |
| Projector - 1st Intermediate Embedding | | | | | | | |
| SIE | 20.1M | 13.07G | 512 | 512 | - | 80.53 | 77.64 |
| CapsIE - 16 | 12.7M | 3.49G | 16 | 256 | - | 74.90 | - |
| CapsIE - 32 | 14.7M | 4.57G | 32 | 512 | - | 78.60 | - |
| CapsIE - 64 | 20.4M | 8.69G | 64 | 1024 | - | 79.24 | - |

projection head ($h_\phi$), we compare various hyperparameterisations, which we describe in the following sections. For primary benchmarking we train our model for 2000 epochs using the Adam Kingma & Ba (2014) optimiser with default settings, a fixed learning rate of 1e-3 and a batch size of 1024. For ablations and sensitivity analyses we train for 500 epochs and employ a batch size of 512, with other settings remaining unchanged. We have found in practice that 500 epochs presents a strong correlation with performance. For all evaluations, pre-training was done with the equivariant criterion optimising for viewpoint rotation transformations. Full details on these transformation groups and the criteria are given in prior sections. By default the objective function weighting are as follows, $\lambda_{inv} = 0.1$, $\lambda_{equi} = 5$, $\lambda_V = 10$, $\lambda_C = 1$. Each self-supervised 2000-epoch pretraining run took approximately 22 hours using three Nvidia A100 80GB GPUs for the 32 capsule model, whereas the 64 capsule models, and required approximately 25 hours using six Nvidia A100 80GB GPUs. For comparison SIE training took approximately 26 hours using three Nvidia A100 80GB GPUs. All evaluation tasks are completed on a single Nvidia A100 80GB GPU and take approximately 6 hours for angle prediction, and 3 hours for classification. Given the computational overheads involved, all results are presented as a single run where the seed is set to "2224" with exception to Table. 4 where the mean and standard deviation of 5 seeds are reported.

## 4.2 Downstream Evaluation

To evaluate the quality of representations learnt under the invariant and viewpoint rotation equivariant self-supervised criterion, we use the standard benchmark approach of learning downstream task-specific networks with frozen representations as input. In our case, we evaluate the representations in three distinct tasks to evaluate both invariant and equivariant properties. We use a linear evaluation training protocol of the frozen representations. Further details of the evaluation protocol are given in the appendix A.3.2.

**Invariant Evaluation.** To evaluate invariant properties of the representation, we train a classifier on either the frozen representations output from the encoder network or the intermediate embeddings of the capsule

network projector. Given our advocacy for CapsNets, we evaluate using both the standard linear classification and a capsule layer whose number of output capsules is set to the number of classes. All methods are trained for 300 epochs by cross entropy.

**Equivariant Evaluation.** Evaluating equivariant properties is achieved through a rotation prediction task in which a three layer MLP is trained to predict the quaternions defining the rotation between two views of the same object. We train for 300 epochs using MSE loss. Similar to rotation prediction, we evaluate the representation's equivariant properties by regressing the colour hue of an object view. We train a single linear layer for 50 epochs using MSE. For evaluating the performance of predicting equivariance, we use the metric $R^2 = 1 - \frac{\sum_i (y_i - \tilde{y}_i)^2}{\sum_i (y_i - \bar{y})^2}$ where $\{y_i\}$ are the ground truth (target values of the equivariance), $\bar{y}$ is the mean value of these targets, and $\{\tilde{y}_i\}$ are the predictions. Higher $R^2$ indicates the model has a better fit for predicting equivariant transformations.

Table 2: **Evaluation of equivariant properties via downstream rotation prediction (*left*) and colour prediction (*right*) tasks.** Representations are learnt under the invariance and rotation equivariant objective, we evaluate both the representations and the intermediate embeddings of the projection head under varying number of capsules. *'-' refers to non-compatible experiments.* For non-capsule models, 'All', 'Inv.', and 'Equi.' refer to the full vector representation, the "left" half representation optimised under the invariant objective, and the "right" half representation optimised under the equivariant objective, respectively. For capsule networks, 'Inv.', and 'Equi.' refer to the activation vector and pose matrices, respectively.

| Method | Rotation Prediction ($R^2$) | | | Colour Prediction ($R^2$) | | |
|---|---|---|---|---|---|---|
| | All | Inv. | Equi. | All | Inv. | Equi. |
| Supervised | | | | | | |
| ResNet-18 | 0.76 | - | - | 0.99 | - | - |
| SR-Caps - 16 | - | - | 0.83 | - | - | 0.99 |
| SR-Caps - 32 | - | - | 0.84 | - | - | 0.99 |
| SR-Caps - 64 | - | - | 0.80 | - | - | 0.99 |
| Encoder Representation | | | | | | |
| SIE | 0.73 | 0.23 | 0.73 | 0.07 | 0.05 | 0.02 |
| CapsIE - 16 | 0.68 | - | - | 0.02 | - | - |
| CapsIE - 32 | **0.74** | - | - | 0.04 | - | - |
| CapsIE - 64 | 0.72 | - | - | 0.01 | - | - |
| Projector - 1st Intermediate Embedding | | | | | | |
| SIE | - | 0.38 | 0.58 | - | 0.45 | 0.09 |
| CapsIE - 16 | - | - | 0.64 | - | - | -0.01 |
| CapsIE - 32 | - | - | **0.71** | - | - | -0.04 |
| CapsIE - 64 | - | - | 0.67 | - | - | -0.09 |

**Representation Quality.** The performance of CapsIE for both invariant and equivariant benchmark tasks is given in Tables 1 and 2, respectively. We evaluate both the representations produced by the ResNet-18 encoder and the intermediate embeddings of the capsule layer projection head given different values for the number of capsules. We observe that across all models that the invariant properties captured within the representations marginally suffer compared to the MLP projector of SIE. This observation is expected given the significantly reduced number of embeddings employed in the invariant criterion compared to SIE. However, the evaluation of equivariant properties captured by the representations demonstrates that the use of a capsule projector in place of an MLP can lead to marginally improved performance in rotation prediction ($\uparrow 0.01\ R^2$) advancing the prior state-of-the-art whist also approaching the supervised baseline.

Additionally, we include a colour prediction task as a diagnostic test to determine whether the capsule projector additionally encodes colour changes, an additional group element, without being explicitly optimised to do so. We hypothesise that poor colour prediction from CapsIE's pose is due to its optimisation being directly for angle and pose changes only. Since we observe these results with $R^2$ slightly below zero. However,

given the negative score is very marginal, around the zero mark, we do not interpret this as a significant negative correlation. Future investigations into the more significant negative correlation of the 64-capsule model, and whether the score negatively correlates with the number of capsules, are warranted. In Table 2, we can conclude that capsules do not inherently encode colour information under such strong objectives. This is beneficial as it allows for more finegrained control for properties that are optimised for and does not arbitrarily encode additional group elements.

**Intermediate Projector Embeddings.** The role of the projector is primarily employed to decorrelate the embeddings on which the objective function operates on from the representations employed downstream. The premise is to avoid representations that are over-fit to the self-supervised objective (Bordes et al., 2023). However, it has been well studied that it can be beneficial to maintain a number of projector layers and instead utilise intermediate projector embeddings for downstream tasks. Specifically, the equivariant information that has been shown to be captured in the object pose (De Sousa Ribeiro et al., 2024).

We evaluate the intermediate embeddings output from the primary capsule layer in the same manner as the representations; however, the activations and pose are given over a spatial region, so we perform average pooling to return an activation vector and a $4 \times 4$ pose for each capsule, which we then flatten into a vector. As with the representation evaluation, we report our invariant and equivariant task performance in Tables 1 and 2, respectively. We find that evaluating the intermediate embeddings of the capsule projector consistently leads to improved performance on rotation-equivariant tasks compared to SIE intermediate embeddings across all settings. In our case, we demonstrate that the preservation of capsule layers for downstream tasks results in significantly better evaluation performance of the representations, suggesting that capsule nets preserve such properties of interest compared to MLPs. Further investigations into this phenomenon are left to future research.

## 4.3 Quantitative Evaluation of Equivariance

To quantitatively evaluate the equivariant performance of our method and capsule projector, we provide evaluations in line with those proposed in Garrido et al. (2023). We report the Mean Reciprocal Rank (MRR) and Hit Rate at k (H@k) on the multi-object setting. Given a source and target pose of an object, we first compute the embeddings of each image and pass the source embedding through the predictor. We then use the resulting vector to retrieve the nearest neighbours.

The MRR is the average reciprocal rank of the target embedding in the retrieved nearest neighbour graph. H@k in this case is computed to be 1 if the target embedding is in the k-NN graph of the predicted embedding, where we only look for nearest neighbours among the views of the same object.

The Prediction Retrieval Error (PRE) gives an evaluation of predictor quality, and is given by the distance between its rotation $q_1 \in \mathbb{H}$ and the target rotation $q_2$ as $d = 1- <q_1, q_2>^2$ of the nearest neighbour of the predicted embedding averaged over the whole dataset.

All the results evaluated by the aforementioned metrics are given in Table 3. Our CapsIE network outperforms EquiMod, Only Equivariance and SIE by a considerable margin across all metrics and for all dataset splits. We achieve strong perfromance on PRE, reporting 0.21 PRE on the validation set compared to 0.48 for EquiMod and Only Equivariance and 0.29 for SIE. The same significant gains in equivariant performance are shown for MRR and H@1 and H@5. Note, a random H@1 results in a performance of 2% (0.02), demonstrating that our method lies well above random.

## 4.4 Number of Capsules

Each capsule, in theory, should represent a unique concept; thus when the number of capsules is increased, logically so should the network's representation ability to capture an increasing number of semantic concepts. Observing the invariant performance during training (Figure 4) and downstream evaluation in Table 1, CapsIE gains a slight improvement with the addition of more capsules. Here, online training refers to a co-optimisation scheme where an evaluation network is trained in parallel during standard pretraining procedure, this allows for analysis and visualisation of training dynamics rather than relying on the losses of each term directly. Since this evaluation network has been trained for far longer and on a much more diverse representation

Table 3: **Quantitative evaluation of the predictor when using a Capsule network projector, using PRE, MRR and H@k.** The source dataset, for which embeddings are computed, and the dataset used for retrieval are given in the format *source-retrieval* for PRE and *source* for MRR and H@k. Here *source* refers to the set from which embeddings are computed (train or val), while *retrieval* corresponds to the set used for comparison/retrieval (train, val, or all = train + val).

| Method | PRE (↓) | | | MRR (↑) | | H@1 (↑) | | H@5 (↑) | |
|---|---|---|---|---|---|---|---|---|---|
| | train-train | val-val | val-all | train | val | train | val | train | val |
| EquiMod | 0.47 | 0.48 | 0.48 | 0.17 | 0.16 | 0.06 | 0.05 | 0.24 | 0.22 |
| Only Equivariance | 0.47 | 0.48 | 0.48 | 0.17 | 0.17 | 0.06 | 0.05 | 0.24 | 0.22 |
| SIE | 0.26 | 0.29 | 0.27 | 0.51 | 0.41 | 0.41 | 0.30 | 0.60 | 0.51 |
| CapsIE | **0.17** | **0.21** | **0.20** | **0.60** | **0.47** | **0.50** | **0.36** | **0.71** | **0.58** |

set during training, the performance is typically higher. The performance however, demonstrates that our model has better utilised the additional representational power to improve performance. This pattern is also observed when evaluating equivariant properties (shown in Figure 4), yet it is less pronounced. However, in Table 1 we also show that increasing the number of capsules in a supervised SR-Caps model trained in a standard supervised fashion is not an indicator of increased performance, aligning with prior capsule research (Everett et al., 2023). This differentiation in behaviour provides an interesting direction for future research.

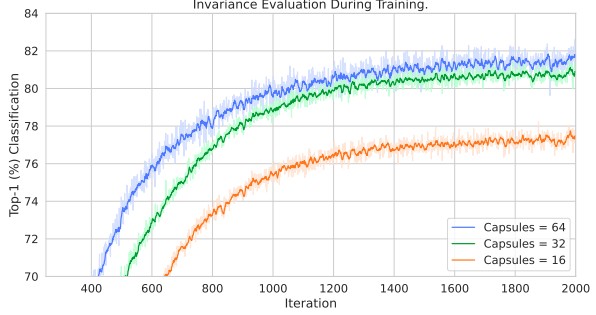 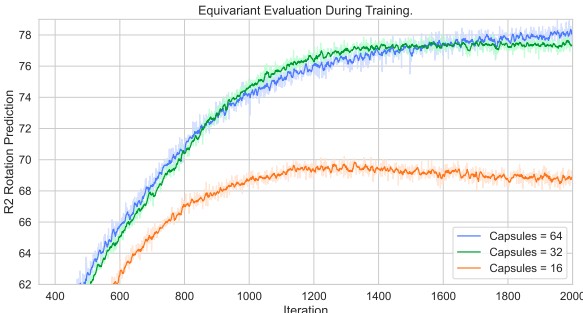

Figure 4: **Invariant and equivariant performance of encoder representations during training when varying the number of capsules.** *(left)* Classification evaluation performance (top-1 %) and *(right)* Rotation prediction online evaluation performance ($R^2$) both learned encoder representations. This evaluation is co-optimised during training and hence differs from the downstream evaluation reported in the Table 1 and 2. We observe that as capsule numbers increase so does performance; however, at low capsule numbers, later in training invariant objective takes precedence.

### 4.5 Additional Quantitative Results

### 4.5.1 Objaverse

Datasets such as 3DIEBench, derived from ShapeNet Chang et al. (2015), have some limitations, considering that they are synthetically generated, although photorealistic. However, datasets such as Objaverse (Deitke et al., 2023) contain real-world scans, therefore providing the possibility of evaluating equivariant models on different settings. To further validate CapsIE on a randomly rotated multi-view dataset, we used a subset of Objaverse-LVIS Deitke et al. (2023) with six classes (airplane, bench, car automobile, chair, coffee table, and gun), following the class selection of Wang et al. (2024). We evaluated classification and rotation prediction using two approaches. Firstly, we perform transfer learning by fine-tuning the entire network pre-trained on 3DIEBench as described previously. Secondly, we take the model pre-trained on 3DIEBench, freeze the backbone encoder, and train only the task-specific heads from scratch following the same evaluation procedure as described in A.3.2.

Table 4: Evaluation on a subset of Objaverse-LVIS using a ResNet-18 backbone. Representations are evaluated on an invariant task (classification) and an equivariant task (rotation prediction). Each model is evaluated five times with different random seeds.

| Method | Pre-training (Frozen Backbone) | | Transfer Learning (Fine-tuning) | |
|---|---|---|---|---|
| | Classification (Top-1) | Rotation ($R^2$) | Classification (Top-1) | Rotation ($R^2$) |
| VICReg | 80.43±1.11 | 0.29±0.008 | 90.22±0.70 | 0.62±0.012 |
| SimCLR | 83.44±0.88 | 0.29±0.003 | 91.08±0.72 | 0.63±0.011 |
| AugSelf | 83.87±0.38 | 0.30±0.015 | 90.75±0.70 | 0.64±0.009 |
| SEN | 82.90±1.17 | 0.30±0.011 | 90.86±0.38 | 0.64±0.015 |
| EquiMod | 83.76±0.45 | 0.30±0.013 | 89.89±0.24 | 0.63±0.011 |
| SIE | 75.27±1.26 | 0.29±0.005 | 89.78±0.66 | 0.62±0.007 |
| CapsIE | 72.58±0.85 | 0.43±0.014 | 90.75±0.45 | 0.64±0.010 |

Table 5: Transfer learning via DETR fine-tuning with frozen backbone on MOVi-E.

| Method | Classification (Top-1) | Rotation ($R^2$) | mAP | mAP$_{50}$ | mAP$_{75}$ |
|---|---|---|---|---|---|
| SIE | 73.7 | 0.20 | 26.47 | 41.83 | 28.26 |
| CapsIE | 72.5 | 0.23 | 28.26 | 43.41 | 31.38 |

Results shown in Table 4 demonstrate that CapsIE performs competitively against other methods and, in some cases, outperforms them by a large margin, e.g. rotation with a frozen backbone. We do note that in the fine-tuning case our capsule architecture outperforms the SIE invariant baseline demonstrating that the learned invariant representations may be generally well expressed but need a further small amount of supervised tuning to better structure the representations more effectively. This introduces a promising direction for future work, and the applicability of pretrained capsule networks.

### 4.5.2 MOVi-E

To assess performance under more realistic and challenging conditions, we evaluate our approach on the Multi-Object Video (MOVi-E) dataset Greff et al. (2022). MOVi-E is a synthetic benchmark containing scenes with up to 17 distinct objects placed in photorealistic environments, often with occlusions and complex backgrounds. Each scene is generated through a 2-second rigid-body simulation in which multiple objects fall and interact. While MOVi-E provides sequences with a linearly moving camera, in our experiments, we sample individual frames and process them independently rather than using the temporal dimension. More information on the dataset can be found in the original paper Greff et al. (2022) and the public repository[1].

In this setting, the task involves detecting every object (via bounding boxes), classifying its type, and estimating its pose relative to the camera frame. For object detection, we adopt the DETR architecture Carion et al. (2020), initialising the ResNet-50 backbone with weights pre-trained on 3DIEBench. The backbone is frozen, while the transformer encoder, decoder, and prediction heads are fine-tuned on MOVi-E. To enable pose estimation, we extend DETR with an additional MLP head that regresses rotation quaternions. This predictor mirrors the bounding box regression module, using a three-layer MLP with 256 hidden units. The quaternion regression loss is defined as mean squared error and added to the overall training objective, which is a weighted sum of individual DETR losses Carion et al. (2020).

Training is carried out for 200 epochs with a batch size of 64, starting from a learning rate of 0.0001 that is reduced tenfold after epoch 100. The quaternion loss is assigned a weight of 2, and the Generalised Intersection over Union loss a weight of 3, while all other hyperparameters remain unchanged from the original DETR setup.

---

[1]https://github.com/google-research/kubric/blob/main/challenges/movi

The results of classification, detection, and rotation regression are summarised in Table 5. Our method outperforms SIE in all tasks except classification, and it is the strongest competing equivariant approach, indicating strong generalisation and robustness to complex multi-object scenarios. Although CapsIE performs competitively, its performance remains constrained by the difficulty of MOVi-E and the fact that the ResNet-50 backbone was trained only in single-object settings. We view this as a promising direction for future work, particularly through adapting the method to exploit temporal video information.

## 5 Related Work

### 5.1 Equivariant Self-Supervised Learning

Self-supervised learning has seen the majority of its success in the invariant setting by contrastive (Chen et al., 2020), information maximisation (Zbontar et al., 2021; Bardes et al., 2022), or clustering methods (Caron et al., 2021; Assran et al., 2022). All families of approaches rely on training a network to be invariant to transformations by increasing the similarity between embeddings of the same image under augmentation. The differing approaches emerge from alternative methods to avoid collapse, a phenomenon where embeddings fall into a lower-dimensional subspace rather than the entire available embedding space, resulting in a trivial solution (Hua et al., 2021). Although these methods differ, they all produce similarly performing representations, hence we employ information maximisation methods as the basis of this work due to their computational efficiency.

Learning to be invariant to transformations is typically useful for semantic discrimination tasks, yet preserving information about the transformations can be highly beneficial. Some approaches have attempted to capture specific information regarding transformations by predicting the applied augmentation parameters (Lee et al., 2021), preserving the strength of augmentations (Xie et al., 2022) and introducing rotational transformations (Dangovski et al., 2022). However, as stated in Garrido et al. (2023), these methods provide no guarantee that a mapping is learnt in the latent space that reflects the transformations in the input space. Hence, methods have been employed that address this limitation (Devillers & Lefort, 2023; Park et al., 2022; Garrido et al., 2023). All of these methods employ predictor networks to predict displacement representations in the latent space, given a single view representation and the transformation parameters. The latter, SIE Garrido et al. (2023), is the basis of our work, which further extends prior methods by splitting representation vectors into invariant and equivariant parts to better separate differing information.

### 5.2 Capsule Networks

CapsNets present an alternative architecture to CNNs, addressing their limitations by explicitly preserving hierarchical spatial relationships between features (Sabour et al., 2017). CapsNets replace scalar neurons with vector or matrix poses, representing specific concepts at different levels of a parse tree as the network goes deeper. The first layer (primary capsules) corresponds to the most basic parts, while capsules in deeper layers represent more complex concepts composed of simpler concepts as they get closer to the final layer, where each capsule corresponds to a specific class.

The key components of the CapsNet are the pose and the activation. The pose of a capsule is an embedding vector or matrix which provides a representation for the concept. The activation scalar is a value between 0 and 1 which represents how certain the network is that the concept is present and can be calculated directly from the values of the pose or via other means via the routing mechanism.

The key novelty in CapsNets is the routing mechanism, which determines the contributions of lower-level capsules to higher-level capsules. Numerous routing algorithms, both iterative and non-iterative, have been proposed to address the efficiency and effectiveness of this process (Sabour et al., 2017; Mazzia et al., 2021; Feng et al., 2024; Ribeiro et al., 2020; Hinton et al., 2018; De Sousa Ribeiro et al., 2020; Yang et al., 2021; Everett et al., 2024). Among these, SRCaps Hahn et al. (2019) introduces a non-iterative routing mechanism. This method retains all the desirable properties of CapsNets, such as equivariance, while largely mitigating the time cost of iterative methods at the expense of a slight performance loss. However, SRCaps faces the same limitations as other CapsNets where high-resolution or high-class datasets are beyond the network's

abilities when trained in a standard fashion. For a more detailed description of capsule routing mechanisms, please check this review De Sousa Ribeiro et al. (2024).

### 5.3 Part-Whole Decomposition and Interpretability

Capsule networks naturally lend themselves to an explicit part–whole decomposition because each capsule encodes an entity's instantiation parameters (pose, presence and related attributes) and lower-level capsules cast "votes" for higher-level capsules so that coherent clusters of votes identify which parts belong to which whole – an idea formalised as routing-by-agreement in the original capsule proposals (Sabour et al., 2017). This voting/routing mechanism produces structured, disentangled representations (pose matrices or vectors plus activation lengths) that are directly interpretable: pose parameters report geometric transformations while activation magnitudes reflect part/object presence, enabling inspection of what parts drove a decision (Hinton et al., 2018; De Sousa Ribeiro et al., 2024). Work on stacked/autoencoding capsule models has further shown that object-level capsules can be learned (even unsupervised) from part poses, strengthening the claim that capsules implement an intrinsic compositional (part→object) inductive bias (Kosiorek et al., 2019). Other formulations that treat routing as posterior inference or introduce explicit routing uncertainty demonstrate how probabilistic/variational routing makes the part–whole assignments and the model's confidence in them explicit, improving robustness and giving a principled way to read out uncertainty about which parts compose a given object (De Sousa Ribeiro et al., 2020).

## 6  Conclusion

Our proposed method demonstrates how self-supervised CapsNets can be employed to better learn equivariant representations, leveraging architectural assumptions, removing the need to explicitly split representation vectors and train separate projector networks. The resulting solution, CapsIE, achieves state-of-the-art performance in equivariant downstream benchmarks with an improvement of 0.01 $R^2$ on prior self-supervised rotation prediction tasks. Our results contribute significantly to the application of CapsNets in self-supervised representation learning, introducing desirable properties with improved effectiveness over MLP projectors.

This work aims to learn higher quality and more applicable representations of images without human-generated annotations; therefore, such methods can lead to positive societal impacts and the development of more accurate or informative models for a number of downstream tasks. However, as is the case with all vision systems, there is potential for exploitation and security concerns; therefore, one should consider AI misuse when extending our method.

As stated in the original dataset proposal and the problem setting, the methodology presented relies on the group elements being known. Hence, the applicability of the proposed method is only possible in settings where group elements are known. We additionally explore alternative equivariant tasks that were not explicitly optimised for as a diagnostic task to evaluate if the capsule projector additionally encodes additional group elements. Our findings determine that our method retains the fine-grained control of learning equivariant embeddings specified by the objective function, a beneficial property presented in prior methods to avoid learning of arbitrary group elements.

While our capsule networks perform well on the equivariant tasks, the invariant performance has suffered as a consequence. We attribute this to the reduced capacity of capsule networks, where the number of embeddings is significantly lower than that of the MLP counterpart. To improve performance, alternative capsule networks other than SRCaps could be employed, which have shown improved classification performance, or the number of capsules can be scaled up, where Figure 4 demonstrated improved top-1 performance with greater capsule number. Lastly, a further limitation of our work is the absence of multiple seed runs in most experiments (with the exception of Table 4), consistent with (Bardes et al., 2022) and (Garrido et al., 2023), due to the disproportionately high computational costs.

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

# A   Appendix

## A.1   3DIEBench Dataset

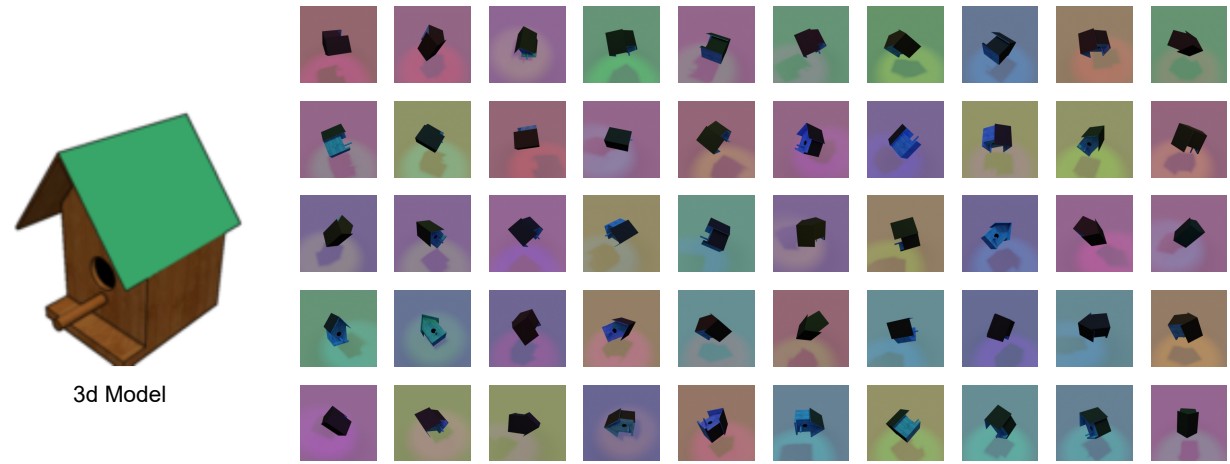

3d Model

Figure 5: **The 3DIEBench dataset**, one 3d model is used to create 50 different views in a synthetic environment, which are saved as images along with the latent values by which they are transformed.

Typical equivariant datasets are generally handcrafted and simple, with a small number of classes and instances within each class. This is due to the time needed in order to ensure correctness. While standard image datasets do allow for testing invariance in the form of augmenting the same image in two different ways, they do not allow for the precise transformation of the subject. Thus, there is a need for a new, synthetic dataset.

We use the 3DIEBench (Garrido et al., 2023) dataset[2], which has been created specifically to be a hard yet controlled test-bed for invariant and equivariant methods. The dataset consists of 52,472 3d objects across 55 classes of 3d objects from ShapeNetCorev2 (Chang et al., 2015) posed in 50 different views as well as the latent information of the view. This can be seen in figure 5. For training we then randomly select two views from each model in the training set. The parameters by which the model could have been augmented are listed in Table 6.

## A.2   3DIEBench Problem Statement

First, we define equivariance by defining a Group consisting of a set $G$ and a binary operation $\cdot$ on $G$, $\cdot : G \times G \to G$ such that $\cdot$ are associative; there is an identity $e$ which satisfies $e \cdot a = a = a \cdot e, \forall a \in G$; and for each $a \in G$ there exists an inverse $a^{-1}$ such that $a \cdot a^{-1} = e = a^{-1} \cdot a$. Group actions are concerned with how groups manipulate sets, where the left group action can be defined as a function $\alpha$ of group $G$ and set $S$, $\alpha : G \times S \to S$ such that $\alpha(e, s) = s, \forall s \in S$, and $\alpha(g, \alpha(h, s)) = \alpha(gh, s), \forall s \in S$, and $\forall g, h \in G$. In our setting, we are concerned with group representations which are linear group actions acting on vector space $V$, which we define as $\rho : G \to GL(V)$ where $GL(V)$ is the general linear group on $V$. Here $\rho(g)$ describes the transformation applied to both the input data $x$ and latent $f(x)$ given parameters $g$ (Park et al., 2022;

---

[2]The full dataset and splits employed can be found at `https://github.com/facebookresearch/SIE`

Table 6: **Values of the factors of variation used for the generation of 3DIEBench.** Each value is sampled uniformly from the given interval. Object rotation is generated as Tait-Bryan angles using extrinsic rotations. Light position is expressed in spherical coordinates. This table is sourced from (Garrido et al., 2023).

| Parameter | Minimum value | Maximum value |
|---|---|---|
| Object rotation X | $-\frac{\pi}{2}$ | $\frac{\pi}{2}$ |
| Object rotation Y | $-\frac{\pi}{2}$ | $\frac{\pi}{2}$ |
| Object rotation Z | $-\frac{\pi}{2}$ | $\frac{\pi}{2}$ |
| Floor hue | $0$ | $1$ |
| Light hue | $0$ | $1$ |
| Light $\theta$ | $0$ | $\frac{\pi}{4}$ |
| Light $\phi$ | $0$ | $2\pi$ |

Garrido et al., 2023). Transformations comprise colour scaling and shifting, and rotations around a fixed point.

Following this, we can define the function $f : X \to Y$ as being equivariant with respect to a group $G$ with representations $\rho_X$ and $\rho_Y$ if $\forall x \in X$, and $\forall g \in G$,

$$f(\rho_X(g) \cdot x) = \rho_Y(g) \cdot f(x). \tag{8}$$

The goal is to therefore learn $f$ and $\rho_Y$ to construct representations that are equivariant to viewpoint transformations when $\rho_X$ is not known, but the group elements $g$ that parameterise the transformations are known.

### A.3  Training protocols

### A.3.1  CapsIE Pre-training

Our proposed CapIE model is comprised of a ResNet-18 encoder, SR-CapsNet comprised of a primary capsule layer routed to a second capsule layer. The SR-CapsNet projector takes as input the activation map output of the ResNet prior to the final global average pooling. The predictor network employed is that described in (Garrido et al., 2023). All details of the architectural design are given in the main paper.

Training of our CapsIE model is done over 2000 epochs with a batch size of 1024, optimised via the Adam optimiser with learning rate 0.001, and default parameters, $\beta_1 = 0.9, \beta_2 = 0.999$. By default the objective function weighting are as follows, $\lambda_{\mathrm{inv}} = 0.1$, $\lambda_{\mathrm{equi}} = 5$, $\lambda_V = 10$, $\lambda_C = 1$ where we empirically found these optimal for our setting. Further performance gains could be achieved by the tuning of such parameters, however, we deemed this unnecessary.

For ablation studies, where we explicitly state, we train for fewer epochs and with a smaller batch size, 500 and 512, respectively. We find in practice that this setting is a strong proxy for full training performance and significantly saves computational resources.

Training time for 2000 epochs with batch size of 1024, as previously stated, took approximately 22 hours using three Nvidia A100 80GB GPUs, with 64 capsule models taking approximately 25 hours using six Nvidia A100 80GB GPUs.

### A.3.2  Downstream Evaluation

In our work, we perform evaluation on both the frozen ResNet-18 representations and the representations from the primary capsules layer, which are evaluated using either a linear classifier or an additional capsule layer acting as class capsules, i.e the number of capsules is set to the number of classes and activations are used as the logits. Here, we detail the exact training protocols to ensure complete reproducability.

For our evaluations we use two different depths of MLP heads, these are: 1. **Deep MLP** referring to an MLP with layers containing in_dim - 1024 - out_dim neurons, with intermediate ReLU activations. 2. **Shallow MLP** referring to a single MLP layer with in_dim in neurons and out_dim out neurons. When we evaluate our primary capsules for angle and colour prediction, we average the 8x8 feature map so that we only have a single pose vector for the entire image. For our Capsule Classification task, we do not have an in_dim as we do not use a MLP, but instead use a capsule layer which operates on the primary capsules pose and activations.

Table 7: **Training settings for our evaluations.** Settings are the same for all number of capsules. **NC** is used as shorthand for number of capsules. - denotes that this element is not used. * denotes multiplication.

| | Representations Angle | Representations Colour | Representations Classification | Capsule Angle | Capsule Colour | Capsule Classification |
|---|---|---|---|---|---|---|
| Caps Head | - | - | - | - | - | Yes |
| MLP Head | Deep | Shallow | - | Deep | Shallow | - |
| in_dim | 512 | 512 | 512 | **NC** * 16 | **NC** * 16 | N/A |
| out_dim | 4 | 2 | 55 | 4 | 2 | 55 |
| Optimizer | Adam | Adam | Adam | Adam | Adam | Adam |
| LR | 0.001 | 0.001 | 0.001 | 0.001 | 0.001 | 0.001 |
| $\beta_1$ | 0.9 | 0.9 | 0.9 | 0.9 | 0.9 | 0.9 |
| $\beta_2$ | 0.999 | 0.999 | 0.999 | 0.999 | 0.999 | 0.999 |
| Batch Size | 256 | 256 | 64 | 256 | 256 | 256 |
| Epochs | 300 | 50 | 300 | 300 | 50 | 300 |
| Objective | MSE | MSE | Cross Entropy | MSE | MSE | Cross Entropy |

### A.3.3 Supervised Training of SR-Caps

In our work we train a Self Routing Capsule Network model in a supervised fashion for the downstream tasks to evaluate whether our pretrained model improves the quality of downstream evaluations. The training setting of these runs can be found in table 8.

Deep MLP refers to an MLP with layers containing number_caps * 16 * 2 - 1024 - 4 neurons, with intermediate ReLU activations. Shallow MLP head refers to a single MLP layer with number_caps * 16 * 2 in neurons and either 4 (for rotation prediction) or 2 (for colour prediction) out neurons.

Table 8: **Training settings for our supervised Self Routing Capsule Network model.** Settings are the same for all numbers of capsules. - denotes that this element is not used.

| | Angle | Colour | Classification |
|---|---|---|---|
| Caps Head | - | - | Yes |
| MLP Head | Deep | Shallow | - |
| Optimizer | Adam | Adam | Adam |
| LR | 0.001 | 0.001 | 0.001 |
| $\beta_1$ | 0.9 | 0.9 | 0.9 |
| $\beta_2$ | 0.999 | 0.999 | 0.999 |
| Batch Size | 256 | 256 | 64 |
| Epochs | 300 | 50 | 300 |
| Objective | MSE | MSE | Cross Entropy |

### A.3.4 Invariant and Equivariant SSL Benchmarks

We report below the classification (invariant, Table 9), rotation prediction, and colour prediction (equivariant, Table 10) performance of baseline self-supervised methods. The below baseline results are acquired from (Garrido et al., 2023), with the exception of those denoted by '*' which corresponds to our re-implementation.

Table 9: **Evaluation of invariant properties on downstream classification task for baseline SSL methods.** We evaluate both the representations and the intermediate embeddings of the projection head when different numbers of capsules in the projection head is used. *'-' refers to non-compatible experiments.*

| Method | Embedding Dims | | Classification (Top-1%) | | |
|---|---|---|---|---|---|
| | Inv. | Equi. | All | Inv. | Equi. |
| Encoder Representation | | | | | |
| VICReg | - | - | 84.74 | - | - |
| VICReg, $g$ kept identical | - | - | 72.81 | - | - |
| SimCLR | - | - | 86.73 | - | - |
| SimCLR, $g$ kept identical | - | - | 71.21 | - | - |
| SimCLR + AugSelf | - | - | 85.11 | - | - |
| EquiMod (Original predictor) | - | - | **87.19** | - | - |
| EquiMod (SIE predictor) | - | - | **87.19** | - | - |
| SIE (Garrido et al., 2023) | 512 | 512 | 82.94 | 82.08 | 80.32 |
| SIE * | 512 | 512 | 82.54 | 82.11 | 80.74 |
| CapsIE - 16 | 16 | 256 | 76.51 | - | - |
| CapsIE - 32 | 32 | 512 | 79.14 | - | - |
| CapsIE - 64 | 64 | 1024 | 79.60 | - | - |
| Capsule Projector - 1st Intermediate Embedding | | | | | |
| SIE | 1024 | 1024 | - | 80.53 | 77.64 |
| CapsIE - 16 | 16 | 256 | - | 74.90 | - |
| CapsIE - 32 | 32 | 512 | - | 78.60 | - |
| CapsIE - 64 | 64 | 1024 | - | 79.24 | - |

Table 10: **Evaluation of equivariant properties on downstream rotation prediction (*left*) and colour prediction (*right*) tasks for baseline SSL methods.** We evaluate both the representations and the intermediate embeddings of the projection head when different numbers of capsules in the projection head is used.

| Method | Rotation Prediction ($R^2$) | | | Colour Prediction ($R^2$) | | |
|---|---|---|---|---|---|---|
| | All | Inv. | Equi. | All | Inv. | Equi. |
| Encoder Representation | | | | | | |
| VICReg | 0.41 | - | - | 0.06 | - | - |
| VICReg, $g$ kept identical | 0.56 | - | - | 0.25 | - | - |
| SimCLR | 0.50 | - | - | 0.30 | - | - |
| SimCLR, $g$ kept identical | 0.54 | - | - | 0.83 | - | - |
| SimCLR + AugSelf | **0.75** | - | - | 0.12 | - | - |
| EquiMod (Original predictor) | 0.47 | - | - | 0.21 | - | - |
| EquiMod (SIE predictor) | 0.60 | - | - | 0.13 | - | - |
| SIE (Garrido et al., 2023) | 0.73 | 0.23 | 0.73 | 0.07 | 0.05 | 0.02 |
| SIE * | 0.72 | 0.21 | 0.71 | 0.06 | 0.05 | 0.03 |
| CapsIE - 16 | 0.68 | - | - | 0.02 | - | - |
| CapsIE - 32 | **0.74** | - | - | 0.04 | - | - |
| CapsIE - 64 | 0.72 | - | - | 0.01 | - | - |
| Projector - 1st Intermediate Embedding | | | | | | |
| SIE | - | 0.38 | 0.58 | - | 0.45 | 0.09 |
| CapsIE - 16 | - | - | 0.64 | - | - | -0.01 |
| CapsIE - 32 | - | - | **0.71** | - | - | -0.04 |
| CapsIE - 64 | - | - | 0.67 | - | - | -0.09 |

