# OpenReview forum: "Capsule Network Projectors are Equivariant and Invariant Learners"
_TMLR — Accepted by TMLR_

### Review · Reviewer_N8YA · 2025-07-03

**Summary Of Contributions:**

The paper presents CapsIE, a self-supervised learning framework that employs Capsule Networks to jointly learn invariant activations and equivariant pose representations, and shows through experiments on the 3DIEBench dataset that this approach achieves better performance on equivariant rotation tasks than prior equivariant self-supervised baselines.

**Audience:**

Yes

**Broader Impact Concerns:**

This paper is purely technological, and does not have ethical issues.

**Claims And Evidence:**

Yes

**Requested Changes:**

1. Please add more explanation or experiments to show when the pose predictor may fail or give non-unique mappings, especially for complex transformations.

2. Clarify how much the use of group theory ideas really helps the final model, or remove extra group theory terms if they are not needed.

3. In the abstract, it is said that "Code is available at anonymised". However, this text does not corresponds to any link to the code. Please modify it.

**Strengths And Weaknesses:**

Strengths

- The paper gives a clear idea to learn both invariant and equivariant features in self-supervised learning.

- It uses Capsule Networks as a projector, which helps to keep part-whole spatial information.

- The method shows better results than some earlier equivariant self-supervised methods on a real dataset.

Weaknesses:

- The paper does not show strong theory to prove that the pose predictor will always learn a good mapping for different transformations. If the mapping is not unique, the training may fail.

- It is not fully clear how the pose vector can really handle very complex or fine-grained changes, like when there are many parts or occlusion. The method may lose detail when objects have complex shapes.

- The method works on a specific dataset with 3D rotations, but there is no test on larger or more diverse scenes. So it is hard to say if the same design can scale well to real-world data with multiple objects and noise.

- The group theory words like “group action” are used, but the model does not really check or use full group properties like closure and inverse. So the theory part may look formal but not very useful in practice.

---

> ### Author Response · Authors · 2025-08-30
> **Responses - Part 1**
>
> Firstly, many thanks for providing excellent feedback and comments. Please find our responses below, as well as a revised manuscript.
>
> 1) **The paper does not show strong theory to prove that the pose predictor will always learn a good mapping for different transformations. If the mapping is not unique, the training may fail.**
>
> The mappings prevent collapse by using regularisation terms on the pose representations. These are therefore enforced to be well distributed, ensuring appropriate diversification in the input and target of the predictor for each transformation. While this does not explicitly exclude the possibility of non-unique mappings, our empirical results of the MMR, PRE, and H@K demonstrate that the mapping is well structured concerning the transformations included in this study.
>
> 2) **It is not fully clear how the pose vector can really handle very complex or fine-grained changes, like when there are many parts or occlusion. The method may lose detail when objects have complex shapes.**
>
> We have provided results for the more complex case where there exists multiple concepts, backgrounds and different models via the Objaverse and MOVi-E datasets. We agree this is a challenge, however, we do see an improvement in our method over prior works. The full exploration of this challenge is beyond the scope of this work, but we will endeavour to address this in the future.
>
> \
> \
>
> **Table 1.**
>
> *Evaluation on a subset of Objaverse-LVIS using a ResNet-18 backbone. Representations are evaluated on an invariant task (classification) and an equivariant task (rotation prediction). Each model is evaluated five times with different random seeds. We report in bold the best performance across all methods.*
>
> | Method  | Pre-training Classification (Top-1) | Pre-training Rotation (R²) | Transfer Classification (Top-1) | Transfer Rotation (R²) |
> | ------- | ----------------------------------- | -------------------------- | ------------------------------- | ---------------------- |
> | VICReg  | 80.43 ± 1.11                        | 29.06 ± 0.79               | 90.22 ± 0.70                    | 62.06 ± 1.17           |
> | SimCLR  | 83.44 ± 0.88                        | 28.80 ± 0.34               | 91.08 ± 0.72                    | 63.36 ± 1.10           |
> | AugSelf | 83.87 ± 0.38                        | 29.80 ± 1.49               | 90.75 ± 0.70                    | 63.69 ± 0.88           |
> | SEN     | 82.90 ± 1.17                        | 29.51 ± 1.14               | 90.86 ± 0.38                    | 63.61 ± 1.52           |
> | EquiMod | 83.76 ± 0.45                    | 29.58 ± 1.25               | 89.89 ± 0.24                    | 62.84 ± 1.09           |
> | SIE     | 75.27 ± 1.26                        | 28.96 ± 0.49               | 89.78 ± 0.66                    | 61.85 ± 0.70           |
> | CapsIE  | 72.58 ± 0.85                        | 43.32 ± 1.39          | 90.75 ± 0.45                    | 63.60 ± 0.95           |
>
> \
> \
>
> **Table 2.**
>
> *Transfer learning via DETR fine-tuning on MOVi-E.*
>
> | Method | Classification (Top-1) | Rotation (R²) | mAP       | mAP₅₀     | mAP₇₅     |
> | ------ | ---------------------- | ------------- | --------- | --------- | --------- |
> | SIE    | 73.7                   | 0.20          | 26.47     | 41.83     | 28.26     |
> | CapsIE | 72.5                   | 0.23      | 28.26 | 43.41 | 31.38 |

---

> ### Author Response · Authors · 2025-08-30
> **Responses - Part 2**
>
> 3) **The method works on a specific dataset with 3D rotations, but there is no test on larger or more diverse scenes. So it is hard to say if the same design can scale well to real-world data with multiple objects and noise.**
>
> Please see above response and tables.
>
> 4) **The group theory words like “group action” are used, but the model does not really check or use full group properties like closure and inverse. So the theory part may look formal but not very useful in practice.**
>
> The formal definition of group actions is to define the problem set and task defined by the dataset task. While our theory implies this is learnt, the proof for capsule networks is not yet understood, hence we reserve this for future work. To help alleviate this concern, we will reserve these definitions to the appendix to avoid confusion.
>
> **Requested Changes:**
>
> 5) **Please add more explanation or experiments to show when the pose predictor may fail or give non-unique mappings, especially for complex transformations.**
>
> We see from our additional MOVi-E evaluations that both the capsule based method and the SIE baseline have trouble in the detection and retrieval of the rotation equivariant parameters. This is one empirical evaluation that demonstrates that both methods are inherently limited when transformations get more complex (in this case rotation, translation, occlusion, texture, backgrounds). This supports the argument made, and leads to some interesting directions for future work, as this is more inherently rooted in self-supervised equivariant learning with predictors, rather than capsule based methods.
>
> 6) **Clarify how much the use of group theory ideas really helps the final model, or remove extra group theory terms if they are not needed.**
>
> We have moved these to the appendix within the dataset definition to avoid confusion.
>
> 7) **In the abstract, it is said that "Code is available at anonymity". However, this text does not correspond to any link to the code. Please modify it.**
>
> We have uploaded the code to the supplementary material for reference during the review process. A link to a github repo will be provided if accepted.

---

> > ### Comment · Reviewer_N8YA · 2025-08-30
> >
> > You mentioned that the code is available in the supplementary material, but I could not locate it. Please clarify where the supplementary material can be accessed, or ensure it is properly uploaded for review.

---

> > > ### Author Response · Authors · 2025-08-31
> > > **Uploaded**
> > >
> > > Apologies we thought it was there but apparently it wasn’t!
> > >
> > > It has now been uploaded as supplementary material.
> > >
> > > Thank you.
> > >
> > > Authors

---

### Review · Reviewer_FDL6 · 2025-07-10

**Summary Of Contributions:**

This paper aims to expand methods for self-supervised learning with an emphasis on creating invariant and equivariant representations. It does so by modifying a past architecture to use Capsule networks. This allows for the joint learning of invariant and equivariant representations without the need for entirely split representations. The CapsIE network performs worse on classification and color prediction than the prior split architecture (SIE). The CapsIE projector performs better than SIE's projector on rotation prediction and related tests of equivariance, while their encoders perform essentially the same.

**Audience:**

Yes

**Claims And Evidence:**

No

**Requested Changes:**

Clarity: The authors should explain the training procedure for the supervised and SIE models better. They should explain the motivation for and interpretation of the color prediction results (were they expecting good color prediction and didn't get it? Or is color meant to be ignored?). Not clear what "co-optimized" means in Figure 4 caption or why the results re different from Table 2. SIE acronym is never actually spelled out. Several typos throughout.

Insufficient evaluations: The results are not presented with standard deviations over multiple random seeds, so there is no way to know if the performance differences are significant. The authors also make this statement "the use of a capsule projector in place of a MLP can lead to vastly improved performance in rotation prediction (↑ 0.01 R2) ". This is obviously false. .01 is by no means a "vast improvement" and it is not even clear if that .01 difference would remain if results from multiple seeds were reported. Any claims of superiority based on the this result should be removed.

Broader goals:  The authors mention that this work can learn high quality representations without human annotation, yet it seems like this class of methodology can only be employed with a dataset containing well annotated pose descriptions (which can only be generated through simulated images or human annotations). And the equivarence learned only applies to exactly the feature the model was trained to be equivariant to (rotation). These facts combined make the problem look increasingly more like a supervised one. Is there a belief that this methodology could somehow be used to learn representations with beneficial features that aren't a direct result of what is trained into them with well-labeled data?

**Strengths And Weaknesses:**

Strengths: the description of the background and problem is clear; the work tackles an established problem in the field; it utilizes a reasonable existing architecture (Capsule Networks) and explores how it could help; the CapsIE models tend to have a lower computational cost

Weakness: there is some clarity and detail lacking in the methods and results; the claimed benefits are not supported by the actual results; not entirely clear how to best use these findings.

---

> ### Author Response · Authors · 2025-08-30
> **Responses**
>
> Firstly, many thanks for providing excellent feedback and comments. Please find our responses below, as well as a revised manuscript.
>
>
> **Requested Changes:**
>
> 1) **Clarity: The authors should explain the training procedure for the supervised and SIE models better. They should explain the motivation for and interpretation of the color prediction results (were they expecting good color prediction and didn't get it? Or is color meant to be ignored?). Not clear what "co-optimized" means in Figure 4 caption or why the results re different from Table 2. SIE acronym is never actually spelled out. Several typos throughout.**
>
> Colour prediction: Colour is included as a diagnostic to test whether the pose additionally encodes colour changes without being explicitly optimised to do so. We were expecting poor colour prediction from CapSIE’s pose since it is optimised to capture angle and pose changes only. Since we observe these results, we can conclude that capsules do not inherently encode colour information under such strong objectives. This is beneficial as it allows for more finegrained control on what properties we are optimising for. We have clarified this within the paper to ensure that the result is not confusing.
>
> Co-Optimised: This term refers to the setting in which an evaluation network is trained in parallel during standard pretraining procedure. This allows for analysis and visualisation of training dynamics rather than relying on the losses of each term directly. Since this evaluation network has been trained for far more time and on a much more diverse representation set the performance is typically higher. However, the standard evaluation procedure reported in tables 1 and 2 is presented as such to ensure fair comparisons.
>
> SIE acronym: we have adjusted on first mention to ensure that SIE is clearly understandable as Split Invariant Equivariant networks.
>
> 2) **Insufficient evaluations: The results are not presented with standard deviations over multiple random seeds, so there is no way to know if the performance differences are significant. The authors also make this statement "the use of a capsule projector in place of a MLP can lead to vastly improved performance in rotation prediction (↑ 0.01 R2) ". This is obviously false. .01 is by no means a "vast improvement" and it is not even clear if that .01 difference would remain if results from multiple seeds were reported. Any claims of superiority based on the this result should be removed.**
>
> We have adjusted our wording to no longer claim it is a vast improvement, but rather a marginal improvement
>
> 3) **Broader goals: The authors mention that this work can learn high quality representations without human annotation, yet it seems like this class of methodology can only be employed with a dataset containing well annotated pose descriptions (which can only be generated through simulated images or human annotations). And the equivarence learned only applies to exactly the feature the model was trained to be equivariant to (rotation). These facts combined make the problem look increasingly more like a supervised one. Is there a belief that this methodology could somehow be used to learn representations with beneficial features that aren't a direct result of what is trained into them with well-labeled data?**
>
> The task and definition of self-supervised learning are upheld in this work as the procedure for generating input and transforming the objects is completely automatic. Here, we employ a blender environment which the user only specifies the models, and the range of transformations, the automatic procedure then provides all input images and annotations. This is akin to other well-established self-supervised methods where augmentations are applied within a range. We simply treat the blender transformation as the augmentation here. Regarding the rotation only equivariance, this is defined by the objective function itself and can be extended to other transformations to cover different action groups. Here we are only experimenting with SO(3), hence rotation. While we agree that this approach is limited when the input is more natural, real-world images, we aim to demonstrate methods that have potential to move into this setting.

---

> > ### Comment · Reviewer_FDL6 · 2025-09-03
> >
> > Thanks for the response. Can the authors please address the issue of comparing only individual models, rather than running multiple random seeds of each?

---

> > > ### Author Response · Authors · 2025-09-03
> > > **Thank you for the follow up**
> > >
> > > Thank you for following up with us. In response to your query:
> > >
> > > Multiple seeds have not been reported for two main reasons:
> > >
> > > a) Comparative works, mainly the ICLR paper VicReg (https://openreview.net/pdf?id=xm6YD62D1Ub) and the ICML SIE (https://proceedings.mlr.press/v202/garrido23b/garrido23b.pdf), do not report random seeds; since these are the main papers we compare against, we deem it essential to follow their exact protocol.
> > >
> > > b) We would need to rerun pretraining and evaluation for all reported algorithms multiple times, including ours, which would unfortunately be too computationally expensive given our available resources for minimal benefits within the timeframe.

---

> > > > ### Comment · Reviewer_FDL6 · 2025-09-05
> > > >
> > > > It is unfortunate that previous works also lack statistical robustness. This is, however, not a reason for poor methodology. If the authors cannot run more random seeds, then please include a statement acknowledging this as a limitation.

---

> > > > > ### Author Response · Authors · 2025-09-05
> > > > >
> > > > > Thank you very much for the suggestion.
> > > > >
> > > > > We recognise this limitation; therefore, we have included a sentence at the end of the conclusion on that matter, as shown in the revised version.

---

### Review · Reviewer_eRzZ · 2025-08-22

**Summary Of Contributions:**

This paper introduces CapsIE, a novel SSL method that uses capsule networks to learn representations that stay consistent in some ways while changing appropriately in others.
Most existing SSL methods use simple NN layers (MLPs) as projection heads, which can be limiting. Previous approaches like SIE had to explicitly split their representations and use separate projection heads to handle this challenge.
CapsIE takes a different approach by using a capsule network as its projection head. This design naturally handles equivariance - claiming it can appropriately adjust representations when the input is viewed from different angles or perspectives, without needing the complex splitting mechanisms used before.
The paper also created a custom loss function designed specifically for their capsule network projector. The authors claimed that this loss function works in two ways: it maintains consistency (invariance) by minimizing entropy, and it ensures appropriate changes (equivariance) by maximizing the similarity between predicted representations and those from augmented views of the same data. They also include a regularization term on the pose embeddings to prevent the model from collapsing into trivial solutions.

**Audience:**

Yes

**Claims And Evidence:**

Yes

**Requested Changes:**

Recommendation to strengthen the work:

* The paper mentioned that CapsNets might preserve additional group elements without prior knowledge, further elaboration on concrete steps to overcome this limitation would strengthen the work.

* While the majority of focus of this paper is on equivariance, a deeper analysis of this trade-off, its practical implications for downstream invariant task and potential mitigation strategy can be good addition to the paper. How does this affect tasks that need invariant representations? What can be done to mitigate any negative effects?

* The authors chose SRCaps due to resource constraints on the 3DIEBench dataset, but this choice hurts performance on invariant properties. The paper needs more discussion or experiments showing how this trade-off impacts overall performance.

* Although the paper identified interesting direction for future research regarding capsule utilization behaviour, it can be more interesting of them offer even preliminary hypothesis or more-in depth discussion on these observations which could enhance the immediate impact of the current submission.

Technical issues for acceptance:

* The paper is lack of objective function hyperparameter(HP) sensitivity analysis (λinv = 0.1, λequi = 5, λV = 10, λC = 1). While a thorough search of finding optimal parameters are resource-intensive, providing some sensitivity analysis or ablation studies on these key weighting parameters can show the robustness of the method and its performance across different HP settings.

* The negative R2 values for the colour prediction Equi are presented without any discussion/explanation in the text

Writing and Organizing issues:
* Figures are placed far from where they're mentioned. Figure 1 is referenced two pages after it appears - either move it closer or provide better descriptions so readers don't have to flip back and forth.
* Figure 3 is never referenced in the text at all.

**Strengths And Weaknesses:**

Strengths:
* The authors claim that CapsIE achieves state-of-the-art performance on rotation tasks using the 3DIEBench dataset, showing a 0.01 R2 improvement over previous self-supervised rotation prediction methods. Their results are getting close to supervised baselines. The quantitative evaluation also confirms significant improvements in PRE, MRR and H@k metrics compared to earlier methods.
* CapsIE is more efficient and uses fewer network parameters than equivalent SSL methods like SIE. The authors attribute this efficiency to their choice of SRCaps.
* The study provides good insights into intermediate embeddings, which leads to better performance on rotation equivariant tasks compared to other SSL methods' intermediate embeddings. This finding further confirms that CapsNets are better at preserving important properties in their layers compared to MLPs.
* Overall, the paper is well-written and easy to understand.

Weaknesses:
* CapsIE is limited by its reliance on known group elements, which restricts its applicability to real-world scenarios where these transformations are often unknown or implicit.
* The results show that CapsIE's ability to capture invariant properties is slightly worse compared to SIE's MLP projector. The authors attribute this to using significantly fewer embeddings for the invariant criterion. While this paper focuses mainly on equivariance, a deeper analysis of this trade-off would be valuable - including its practical implications for downstream invariant tasks and potential solutions.
* While SRCaps is efficient, it comes at the cost of slightly lower classification accuracy compared to other capsule architectures.

---

> ### Author Response · Authors · 2025-08-30
> **Responses - Part 1**
>
> Firstly, many thanks for providing excellent feedback and comments. Please find our responses below, as well as a revised manuscript.
>
>
> 1) **CapsIE is limited by its reliance on known group elements, which restricts its applicability to real-world scenarios where these transformations are often unknown or implicit.**
>
> This is a known weakness and limitation of the current state of equivariant self-supervised learning and applies to all methods compared against. However, the task and definition of self-supervised learning are upheld in this work as the procedure for generating input and transforming the objects is completely automatic. The 3DIEBench dataset employs a blender environment in which the user only specifies the models, and the range of transformations, the automatic procedure then provides all input images and annotations. This is akin to other well established self-supervised methods where augmentations are applied within a range. We simply treat the blender transformation as the augmentation here. While we agree that this approach is limited when the input is more natural, real-world images, we aim to demonstrate methods that have potential to move into this setting.
>
> 2) **The results show that CapsIE's ability to capture invariant properties is slightly worse compared to SIE's MLP projector. The authors attribute this to using significantly fewer embeddings for the invariant criterion. While this paper focuses mainly on equivariance, a deeper analysis of this trade-off would be valuable - including its practical implications for downstream invariant tasks and potential solutions.**
>
> The practical limitations regarding the performance of object classification are given which is an immediate measure of downstream performance. To address this, more embeddings could be employed as we state to increase capacity. The hyperparameters could be further tuned for more invariant performance, or an alternative capsule network could be employed. The latter we aim to provide results for by the end of the discussion period.
>
> 3) **While SRCaps is efficient, it comes at the cost of slightly lower classification accuracy compared to other capsule architectures.**
>
> We agree with this raised weakness, SRCaps was used for its computational efficiency not performance. However, our network is agnostic to capsule architectures with separate pose and activation representations.
>
> **Requested Changes:
> Recommendation to strengthen the work:**
>
> 4) **The paper mentioned that CapsNets might preserve additional group elements without prior knowledge, further elaboration on concrete steps to overcome this limitation would strengthen the work.**
>
> We have elaborated in the text, that this limitation is debunked by the color prediction performance where this evaluation is included as a diagnostic to test whether the pose additionally encodes colour changes without being explicitly optimised to do so. We were expecting poor colour prediction from CapSIE’s pose since it is optimised to capture angle and pose changes only. Since we observe these results we can conclude that capsules do not inherently encode colour information under such strong objectives.
>
> 5) **While the majority of focus of this paper is on equivariance, a deeper analysis of this trade-off, its practical implications for downstream invariant task and potential mitigation strategy can be good addition to the paper. How does this affect tasks that need invariant representations? What can be done to mitigate any negative effects?**
>
> To help address this concern, we provide results of our method and comparisons on two new datasets (Objaverse and MOVi-E) which introduce more complex settings of transformations, textures and occlusions, as well as introducing new invariant tasks. Regarding the invariant tasks we find that our capsule network while still performing lower in classification within single object and multi-object settings is still comparable with the SIE approach. Thus demonstrating that the slight decline in performance is at least repeatable. Positively the equivariant rotation task and object detection do continue to outperform the SIE baseline.
>
> We do note that in the fine-tuning case with Objaverse our capsule architecture outperforms the SIE baseline demonstrating that the learned invariant representations may be generally well expressed but need a further small amount of supervised tuning to better structure the representations. This is an interesting finding that we are keen to explore in future work.

---

> > ### Author Response · Authors · 2025-08-30
> > **Responses - Part 2**
> >
> > 6) **The authors chose SRCaps due to resource constraints on the 3DIEBench dataset, but this choice hurts performance on invariant properties. The paper needs more discussion or experiments showing how this trade-off impacts overall performance.**
> >
> > We think that it would be interesting followup work to try different capsule network architectures or even design a capsule network routing algorithm that would specifically excel in this framework. But due to VRAM and compute limitations we could not at the time experiment with other architectures. We have made this potential line of future work explicit in the conclusion, and have made it clear when we introduce our architecture why SR Caps was chosen. However, our network is agnostic to capsule architectures with separate pose and activation representations.
> >
> > 7) **Although the paper identified interesting direction for future research regarding capsule utilization behaviour, it can be more interesting of them offer even preliminary hypothesis or more-in depth discussion on these observations which could enhance the immediate impact of the current submission.**
> >
> > Great suggestion. We have added a subsection (5.3) that discusses these aspects.
> >
> > 8) **The paper is lack of objective function hyperparameter(HP) sensitivity analysis (λinv = 0.1, λequi = 5, λV = 10, λC = 1). While a thorough search of finding optimal parameters are resource-intensive, providing some sensitivity analysis or ablation studies on these key weighting parameters can show the robustness of the method and its performance across different HP settings.**
> >
> > Our choice of objective hyperparameters is grounded by the choices from SIE which itself are taken from the original VICReg paper. We have kept them the same where possible for fair comparisons.
> >
> > 9) **The negative R2 values for the colour prediction Equi are presented without any discussion/explanation in the text**
> >
> > We have also discussed this with the other reviewers and have amended the text. Thank you for raising this.
> >
> > 10) **Figures are placed far from where they're mentioned. Figure 1 is referenced two pages after it appears -  either move it closer or provide better descriptions so readers don't have to flip back and forth. Figure 3 is never referenced in the text at all.**
> >
> > We have reformatted the positioning of the figures to better match where they are discussed and also ensured that figure 3 is referenced in the correct place in the text.

---

### Author Response · Authors · 2025-08-30
**Thank you and revised manuscript**

Dear All,

We would first like to thank all reviewers for their constructive feedback, which has been taken into account to improve our paper. We believe we have addressed all concerns and have also uploaded a revised manuscript with all changes made.
We have also responded to each reviewer's comments individually.

Many thanks again.

Regards,

Authors

---

### Decision · Action_Editor_8kJk · 2025-09-27

**Recommendation:** Accept with minor revision

**Additional Comments:**

Required minor fixes:
1. Make loss definitions consistent (Sec. 3.3).
Align the text and formulas: either (a) keep Eq. (6) and say you use an L2 objective, or (b) change Eq. (6) to a cosine objective; clarify the sign/role of the mean‑entropy regularizer in Eq. (5) for a minimization objective.
2. Fix the rotation parameterization statement.
Replace “g\in\mathbb{R}^3 corresponds to the quaternions” with a correct description (e.g., g are Tait–Bryan angles in \mathbb{R}^3; quaternions are used only for evaluation/prediction), and cite Appendix Table 6. (pp. 4, 15.)
3. Unify the R² reporting scale.
Use the same [0, 1] scale across all tables or add an explicit “×100” note to Table 4; ensure the caption states the convention. (p. 10 vs. p. 7.)
4. Clarify MOVi‑E fine‑tuning phrasing.
The section title says “DETR fine‑tuning,” while the text freezes the ResNet‑50 backbone and fine‑tunes the heads. Please standardize the wording (e.g., “fine‑tuning with frozen backbone”). (p. 10.)
5. Harmonize the conclusion with the color‑prediction diagnostic.
The conclusion currently suggests “additional group elements are preserved without prior knowledge,” yet Table 2 shows near‑zero/negative R² for color prediction (i.e., color is not encoded under the strong pose objective). Please rephrase to avoid over‑generalization and to match the diagnostic’s interpretation. (pp. 7, 12.)
6. Be explicit about seeds where the main results appear.
You acknowledged in the discussion that most 3DIEBench results are single‑run (except Table 4). Please mirror that caveat near Tables 1–3 (caption or §4.1) so readers see it alongside the results. This was requested in review and acknowledged by the authors.
7. Code availability.
The abstract still says “Code will be released upon acceptance,” whereas the OpenReview thread later notes a supplementary upload. Please provide a permanent repository link in the camera‑ready.
8. (Small) efficiency context.
Table 1 reports Params/FLOPs, and §4.1 gives training time for CapsIE, but there is no matching time/memory figure for SIE. A one‑line note (even approximate) would make the efficiency claim more concrete. (pp. 5–6.)

**Audience:**

Yes

**Audience Explanation:**

The work sits squarely in the active area of invariant/equivariant self‑supervised learning, introduces a capsule projector in place of an MLP, and provides competitive cross‑dataset evidence. The efficiency angle and the analysis of intermediate capsule embeddings will interest readers working on representation structure and inductive biases.

**Claims And Evidence:**

Yes

**Claims Explanation:**

On the core claim that a capsule projector can jointly learn invariant activations and equivariant poses and improve equivariant downstream performance, the evidence is persuasive:
- On 3DIEBench, the method matches or very slightly exceeds SIE on rotation prediction, while clearly improving predictor-based retrieval metrics PRE/MRR/H@k. The paper’s wording was toned down from “vast” to “marginal” improvement in response to review, which better reflects the effect size.
- Added experiments on Objaverse and MOVi‑E extend beyond synthetic single-object scenes: CapsIE shows strong rotation R² when the backbone is frozen on Objaverse and outperforms SIE on MOVi‑E detection and rotation, supporting external validity for equivariant tasks.

At the same time, a few aspects reduce statistical and presentational clarity:
- Statistical robustness. Objaverse results report 5 seeds (Table 4), but the main 3DIEBench tables (1–3) remain single-run; this was flagged by a reviewer, acknowledged by the authors, and noted as a limitation only in the conclusion. It would be cleaner if the limitation were also stated where the main results are shown.
- Objective notation. The text says the equivariant head maximizes cosine similarity, but Eq. (6) uses an L2 loss; likewise, the invariant term is described as mean-entropy maximization, yet Eq. (5) adds H(\bar Z) with a positive sign in a minimization objective. The narrative and equations should be made consistent.
- Rotation parameterization. The method section states “g\in\mathbb{R}^3 corresponds to the quaternions of the rotation,” but quaternions are 4‑D. Appendix Table 6 clarifies rotations are generated via Tait–Bryan angles. This discrepancy should be fixed.
- Metric scale. R² is on [0, 1] in Tables 1–3, but Table 4 reports numbers like 43.32, which reads as “×100”; the paper should standardize the scale or annotate clearly.
Overall, the empirical case for the main contribution is solid; the above edits would resolve the remaining clarity issues.